# GDNF receptor GFRα1 is necessary for the maintenance of dopaminergic neurons in the adult substantia nigra

Alvaro Carrier-Ruiz[1]*, Annika Andersson[1¤a], Diana Fernández-Suárez[1¤b], Carlos F. Ibáñez[1,2,3]

1 Department of Neuroscience, Karolinska Institutet, Stockholm, Sweden, 2 Peking University School of Life Sciences, Peking-Tsinghua Center for Life Sciences, PKU-IDG/McGovern Institute for Brain Research, Beijing, China, 3 Chinese Institute for Brain Research, Life Science Park, Beijing, China

¤a Present address: Department of Women's and Children's Health, Karolinska Institutet, Stockholm, Sweden
¤b Present address: Veranex, Gothenburg, Sweden
* alvaro.carrier.ruiz@ki.se

## Abstract

GFRα1 and Ret are the two necessary components of the receptor for GDNF, a neurotrophic factor discovered in the early 1990's for its ability to support the survival of midbrain dopaminergic neurons, including those in the substantia nigra (SN) that project to the dorsal striatum (dSTR) and degenerate in Parkinson's Disease. Several GDNF clinical trials have been conducted to date with mixed results. Despite the physiological and clinical importance of this signaling system, whether any of its components are required for the maintenance of adult SN neurons has not yet been elucidated. In this study, we first analyzed postnatal expression patterns of *Gfra1* and *Ret* in the SN and established that mRNA levels peak at mouse postnatal day 15 (P15), stabilizing after P30. Using Tamoxifen-induced deletion of *Gfra1* at 3 months of age, we found that GFRα1 is required for the maintenance of a subset of adult SN dopaminergic neurons. FluoroGold tracing of SN axons from the dSTR in mutant mice revealed that ablation of GFRα1 preferentially affects the subset of GFRα1-expressing neurons that project to the STR. In addition to the well-known neuroprotective functions of GDNF/GFRα1/RET signaling, our results establish a physiological requirement of the GFRα1 component of this neurotrophic system for the continuous maintenance of SN dopaminergic neurons in the adult brain.

## Introduction

GDNF (glial cell line-derived neurotrophic factor) was first identified in the supernatant of the rat B49 glial cell line by its ability to support the survival of ventral midbrain dopaminergic neurons *in vitro* [1]. Subsequent studies identified three additional genes encoding proteins with sequence similarly to GDNF, thus defining

**Data availability statement:** All relevant data are within the manuscript and its Supporting Information files.

**Funding:** C. F. I. Grant number: 2020-01923 Swedish Research Council https://www.vr.se/english.html. The funders did not play any role in the study design, data collection and analysis, decision to publish, or preparation of the manuscript.

**Competing interests:** The authors have declared that no competing interests exist.

a four-member family of neurotrophic molecules distinct from the at the time much better known neurotrophins (reviewed in [2]). GDNF and its siblings display many of the characteristic functions of neurotrophic proteins, including the ability to promote neuronal differentiation, migration, and survival (reviewed in [2]). Although GDNF shows structural similarities to members of the transforming growth factor beta (TGF-β) superfamily, its receptors are distinct from both TGF-β ligands and neurotrophins. While the RET receptor tyrosine kinase functions as the main signalling subunit [3,4], it requires a glycosyl-phosphatidylinositol (GPI) anchored subunit known as GDNF Family Receptor alpha (GFRα) for efficient ligand binding and activation [5,6]. The mammalian genome encodes four distinct GFRα proteins (GFRα1–4) showing specific affinities for different members of the GDNF family (reviewed in [7]). An alternative receptor system, still based on GFRα proteins, utilizes the neural cell adhesion molecule NCAM instead of RET to promote neurite outgrowth and neuronal migration in different areas of the nervous system [8–12]. The glycoprotein Syndecan-3 has also been proposed as a receptor for GDNF family ligands [13], but its possible roles *in vivo* remain to be substantiated. Similar to other GPI-anchored proteins, GFRα receptors can be released from the membrane by phospholipase-mediated cleavage, allowing them to function in a soluble state and present GDNF ligands to RET on nearby cells [14]. This mechanism, known as *trans*-signaling, contributes to diverse biological processes including axonal guidance, lymphoid tissue morphogenesis and cell migration (reviewed in [15]). It also accounts for the distinct expression patterns of RET and GFRα receptors in the nervous system, where many GDNF-responsive neurons expressing RET project to brain regions rich in GFRα1 [16,17].

The ability of GDNF to promote the survival of midbrain dopaminergic neurons made it a prime candidate for therapeutic approaches to Parkinson's disease (PD), where a large fraction of these neurons in the substantia nigra (SN) are lost to degeneration. Despite numerous clinical trials, however, it remains unclear whether GDNF will ever become a treatment for this disease as the results have so far been very varied and overall inconclusive (reviewed in [18]). At the time of this writing, there are still ongoing efforts using a number of different delivery systems to test GDNF clinically in PD patients. The presence of functional GDNF receptors in dopaminergic neurons and the expression of GDNF in the target areas of these cells in the striatum (STR) had initially fueled the notion that this ligand functions as a target-derived signal to promote the survival, projection and maintenance of SN neurons [16,17]. However, early work directed towards elucidating the physiological function of endogenous GDNF and its receptors was hampered by the early lethality of mice lacking any of these components from birth, as it transpired that they are all necessary for kidney development and enteric nervous system survival (reviewed in [2,7]). Subsequent genetic studies using conditional ablation of GDNF expression in mice have presented conflicting results, with some arguing for and some against a role for endogenous GDNF in the survival and maintenance of dopaminergic neurons in the adult brain [19–21]. On the other hand, while conditional ablation of RET did not result in deficits in SN dopaminergic neurons up to 9 months of age [22], up to 30% dopaminergic neuron loss was observed in 1 and 2 years old conditional mutant mice

[23], suggesting that endogenous RET signaling may not be required initially, but subsequently for the long term maintenance of a subset of SN dopaminergic neurons. Even less information is available on the physiological roles of GFRa1 in the adult nigro-striatal system. Using a conditional allele of the *Gfra1* gene, our team demonstrated functions for this receptor in the survival of cerebellar molecular layer interneurons, Purkinje cell migration and synaptic stability and function of adult medial habenula neurons [8,24,25].

In this study, we re-examined the developmental expression of *Gfra1* and *Ret* in the postnatal SN and dorsal striatum (dSTR), characterizing the neuronal populations that express these receptors during the first 2 months of age. Importantly, we also report on a previously unknown and essential role for GFRα1 in the adult nigro-striatal circuit as evidenced by a selective reduction of dopaminergic neurons projecting to the dSTR and decreased striatal innervation following acute loss of GFRα1 in SN neurons.

## Materials and methods

### Mice

Mice were group housed, two to five per cage, under standard conditions in a temperature (23°C) and humidity (55%) controlled environment, on a 13/11-hour light/dark cycle (including 1 hour of dawn and dusk) with access to food and water *ad libitum*. All surgical procedures and experiments were performed during the light cycle using mice habituated with the experimenter's handling in order to minimize the procedure's stress. Surgical procedures were always carried under isofluorane anesthesia with a combination of buprenorphine and meloxicam administered pre- and post-operative to alleviate suffering, according to recommendations from the animal facility veterinary. The following transgenic mouse lines were used for experiments: Gfra1$^{CreERT2}$ [8,24,25], Gfra1$^{flox}$ (kindly provided by J.O Andressoo and M. Saarma, University of Helsinki), and Rosa26$^{dTOM}$ [26]. All animals were bred in a C57BL6/J background (Charles River, Netherlands) except from Gfra1$^{flox}$, which was kept in a CD1 background (Charles River, Netherlands). Therefore, all animals resulting from breeding Gfra1$^{CreERT2}$ and Gfra1$^{flox}$ mice were ~50:50 C57BL6/J: CD1 background. Wild-type C57BL6/J mice used in this study were purchased from Charles River, Netherlands. Males and females were both used in this study without any experimental sex differences being detected. Animals of age P15 and older were always euthanized by transcardial perfusion under profound isofluorane aneasthesia, whereas P5 and younger mice were quickly decapidaded to avoid any suffering. Animal protocols were approved by Stockholms Djurförsöksetiska Nämnd and are in accordance with the ethical guidelines of Karolinska Institutet (ethical permits no. 21838-2021 and 10377-2022), following EU directive 2010/63/EU and the Swedish Board of Agriculture's regulation and general advice on research animals SJVFS 2019:9 saknr L150.

### Tamoxifen administration

Activation of the CreERT2 system was achieved by tamoxifen induction. Three-month old adult mice were given a daily intraperitoneal (i.p.) injection with 100 mg/kg of tamoxifen (Sigma-Aldrich) dissolved in 10% ethanol (EtOH) in corn oil (Sigma-Aldrich), for four consecutive days. Three days after the last injection, mice were prepared for histological analysis.

### Stereotaxic surgery and retrograde tracing

Three-month old adult mice were anesthetized by isofluorane inhalation (1% to 5%, Baxter Medical, Sweden) and placed on a stereotaxic frame (Stoelting, IL, USA) with a warm heat pad that kept the body temperature at 37°C. Then, the region of the skull above the right dorsal striatum (dSTR) was surgically exposed, and a hole was drilled. A total of 300 nL of 4% FluoroGold (FLG, Fluorochrome LLC, CO, USA) in 0.9% NaCl solution (Braun, Germany) was injected unilaterally in the right dSTR at the following coordinate obtained from the Paxinos and Franklin mouse brain atlas: AP + 1.42 mm, ML + 1.40 mm, and DV −3.2 mm. Injections were performed with a Wiretrol capillary micropipette (Drummond Scientific Company, PA, USA) by nanoliter pressure injection at a flow rate of 100 nL per min (Quintessential Stereotaxic Injector, Stoelting, IL, USA). The pipette was left in place for 5 minutes after the injection before retracting it slowly from the brain to

avoid FLG spillover. The incision was then sutured, and mice were returned to their home cages. One week after surgery, mice were subjected to the tamoxifen administration regime described above.

## Tissue processing

Mice were deeply anesthetized with isoflurane and transcardially perfused with 20 ml of 0.125 M phosphate buffered saline (PBS, pH 7.4, Gibco, United Kingdom) and 40 ml of 4% paraformaldehyde (PFA, Histolab Products, Sweden). Brains were removed, post-fixed in 4% PFA overnight, and cryoprotected in 30% sucrose in PBS. Mice analyzed at postnatal (P) ages P0 and P5 were instead swiftly sacrificed by decapitation, and the dissected heads were immersed in 4% PFA for 48h before cryoprotection. Afterwards, brains were dissected and sectioned in a microtome (Leica SM2000 R, ThermoFisher Scientific, Germany) surrounded by dry ice to obtain 30-µm free-floating coronal slices. These sections were stored at −20°C in a cryoprotective solution containing 1% DMSO (Sigma-Aldrich) and 20% glycerol (Sigma-Aldrich) in PBS. Alternatively, brains for *in situ* hybridization were embedded in OCT cryomount (Histolab Products, Sweden) after cryoprotection, frozen at −80°C, coronally cut with a cryostat (CryStarNX70, ThermoFisher Scientific), mounted on Superfrost Plus microscope slides (ThermoFisher Scientific) and directly stored at −20°C.

## *In situ* hybridization

*In situ* hybridization was performed in mounted 30-µm coronal sections containing the SN or the dSTR from P0, P5, P15, P30 and P60 C57BL6/J mice using the RNAscope Multiplex Fluorescent Reagent Kit v2 (Advanced Cell Diagnostics Biotechne, USA), following the manufacturer's protocol. Briefly, sections were dehydrated and underwent pretreatment to block endogenous peroxidase activity and to optimally permeabilize the samples. Target probes were hybridized and visualized using TSA Vivid Fluorophore kits (Advanced Cell Diagnostics Biotechne, USA). After *in situ* hybridization, samples were immediately processed for immunohistochemistry starting from the blocking step. The following probes and dyes were used in this study: GFRa1 (Mm-Gfra1-C2, Cat No. 431781-C2); Ret (Mm-Ret-C3, Cat No. 431791-C3); TSA Vivid Fluorophore Kit 520, 570 and 650 (Cat. No. 7523, 7626 and 7527).

## Immunohistochemistry

Free-floating brain sections were thawed and washed (3x, 5 minutes each) with washing solution containing 0.3% Triton X-100 (Sigma-Aldrich) in PBS. They were then incubated for 1 hour with a blocking solution made of 5% normal donkey serum (NDS, Jackson Immunoresearch) and 0.3% Triton X-100 in PBS. Primary antibodies were applied overnight at 4°C in blocking solution. After incubation, sections were washed (3x, 20 minutes each) and incubated in a PBS solution with the corresponding secondary antibodies plus 0.1 mg/ml of 40-6-diamidino-2-phenylindole (DAPI, Sigma-Aldrich) for 2 hours at room temperature (RT). Slices from retrograde-traced mice were not counterstained with DAPI because it has a similar excitation/emission spectrum as FLG. Finally, sections were washed with PBS (3x, 30 minutes each), mounted on Superfrost Plus microscope slides, dried and coverslipped in fluorescent mounting medium (S302380-2, Agilent, Dako). Samples that had previously undergone *in situ* hybridization were already mounted. The antibodies used in this study were the following: rabbit polyclonal anti-TH (1:1000, ab152, Sigma-Aldrich); rabbit polyclonal anti-Darpp-32 (1:200, sc-11365, Santa Cruz Biotech) and donkey anti-rabbit IgG (H + L) Alexa Fluor 488, 555 and 647 (1:1000, Cat. No. A-21206, A-31572 and A-31573, ThermoFisher Scientific).

## Image acquisition and analysis

All fluorescent images were captured with a Carl Zeiss LSM 710 confocal microscope with ZEN 2011 software (Carl Zeiss, Germany). All images from the same brain area and experiment were obtained using the identical settings. Image analysis

was made with ImageJ and Fiji 1.54f software (National Institutes of Health). Quantification of the number of RNA puncta was performed according to the manufacturer's recommendation in ImageJ, excluding particles smaller than 0.02 µm$^2$. For each experiment, images from both hemispheres were taken, except for retrograde tracing with FLG, which was performed unilaterally. In FLG traced mice, images from the anterior and posterior (close to the VTA) SN were done to take into account the spread of FLG into the STR.

## Statistical analysis

Statistics analyses were performed using GraphPad Prism 6 (GraphPad Software, Inc.). Data are expressed as the mean±standard error of the means (SEM). Sample sizes were not predetermined statistically but were based on prior studies from our laboratory and standard field practices. Group comparisons were made using an unpaired Student's t-test or one-way analysis of variance (ANOVA), followed by Tukey's multiple comparison test when appropriate. Differences were considered statistically significant when $p < 0.05$.

## Results and discussion

### Developmental expression of *Gfra1* mRNA in the postnatal SN and dSTR

Postnatal mRNA expression levels of *Gfra1* and *Ret* in the mouse SN were assessed by RNAscope (Fig 1A–1E and 1A'–1E'). Dopaminergic neurons were identified by immunohistochemistry for tyrosine hydroxylate (TH) (Fig 1A"–1E"). Quantification across replicate brains of wild type C57BL6 mice revealed a peak in mRNA expression at postnatal (P) day 15 which then stabilized at about half of peak levels from P30 onwards for both mRNA species (Fig 1F and 1G). The peak and subsequent decline in *Gfra1* and *Ret* mRNA expression coincides with the approximate timing of dopaminergic neuron cell death in this brain structure as established in previous studies [27], and suggest that the decrease observed in cells expressing *Gfra1* and *Ret* mRNAs may be due to programmed cell death. Sparse signal for *Gfra1* mRNA was also found in the dSTR (Fig 2A–2E). Medium spiny neurons, the main targets of SN dopaminergic innervation in the dSTR, were identified by immunohistochemistry for Darpp-32 (Fig 2A'–2E'). A peak in *Gfra1* mRNA expression in the dSTR was detected at P30, about 2 weeks later than in the SN (Fig 2F), suggesting that this mRNA signal may have originated from incoming SN dopaminergic axons. Indeed, using a dTomato reporter (which predominantly stains cell bodies) expressed from the *Rosa26* locus under the control of *Gfra1*^CreERT2 (*Gfra1*^CreERT2/+;*Rosa26*^dTOM), it could be confirmed that GFRα1-expressing dTOM+ cells are abundantly present in the lateral septum (LS) at 3 months of age and sparsely in the ventral striatum (vSTR) but not in the dSTR (Fig 2G and 2G'). A previous study found weak signals for *Gfra1* mRNA in striatal cells using conventional *in situ* hybridization [28]. Based on the absence of dTOM+ cell bodies in dSTR, we believe that the sparse *Gfra1* mRNA signal detected in this area originated from the axons of dopaminergic SN neurons.

### GFRα1 is necessary for the maintenance of dopaminergic neurons in the adult SN

The impact of acute deletion of *Gfra1* on adult SN dopaminergic neurons was investigated by comparing the number of dTOM+ cells (reflecting GFRα1+ cells) and TH+ cells in the SN of heterozygous *Gfra1*^CreERT2/+;*Rosa26*^dTOM mutant mice (denoted GFRα1^+/-, Fig 3A and 3A") and null *Gfra1*^CreERT2/flox;*Rosa26*^dTOM mutants (GFRα1^-/-, Fig 3B and 3B") with wild type mice (GFRα1^+/+, Fig 3C and 3C') injected with tamoxifen at 3 months of age. We note that insertion of CreERT2 in the *Gfra1* locus inactivates the targeted allele, rendering *Gfra1*^CreERT2/+ mice heterozygous with regards to GFRα1 expression [8,24,25]. Upon tamoxifen administration in *Gfra1*^CreERT2/flox;*Rosa26*^dTOM mice, Cre-mediated recombination abolishes GFRα1 expression from the second (flox) allele in all cells with an active *Gfra1* locus, rendering such mice homozygous null with regards to GFRα1 expression. At the same time, these cells become labeled by dTOM which reports the activity of the *Gfra1* locus. A 40% reduction in the number of dTOM+ cells in the SN of null mutants compared to heterozygous mice was observed 3 days after the last tamoxifen injection (Fig 3D), indicating an acute dependence on GFRα1 expression for the maintenance of

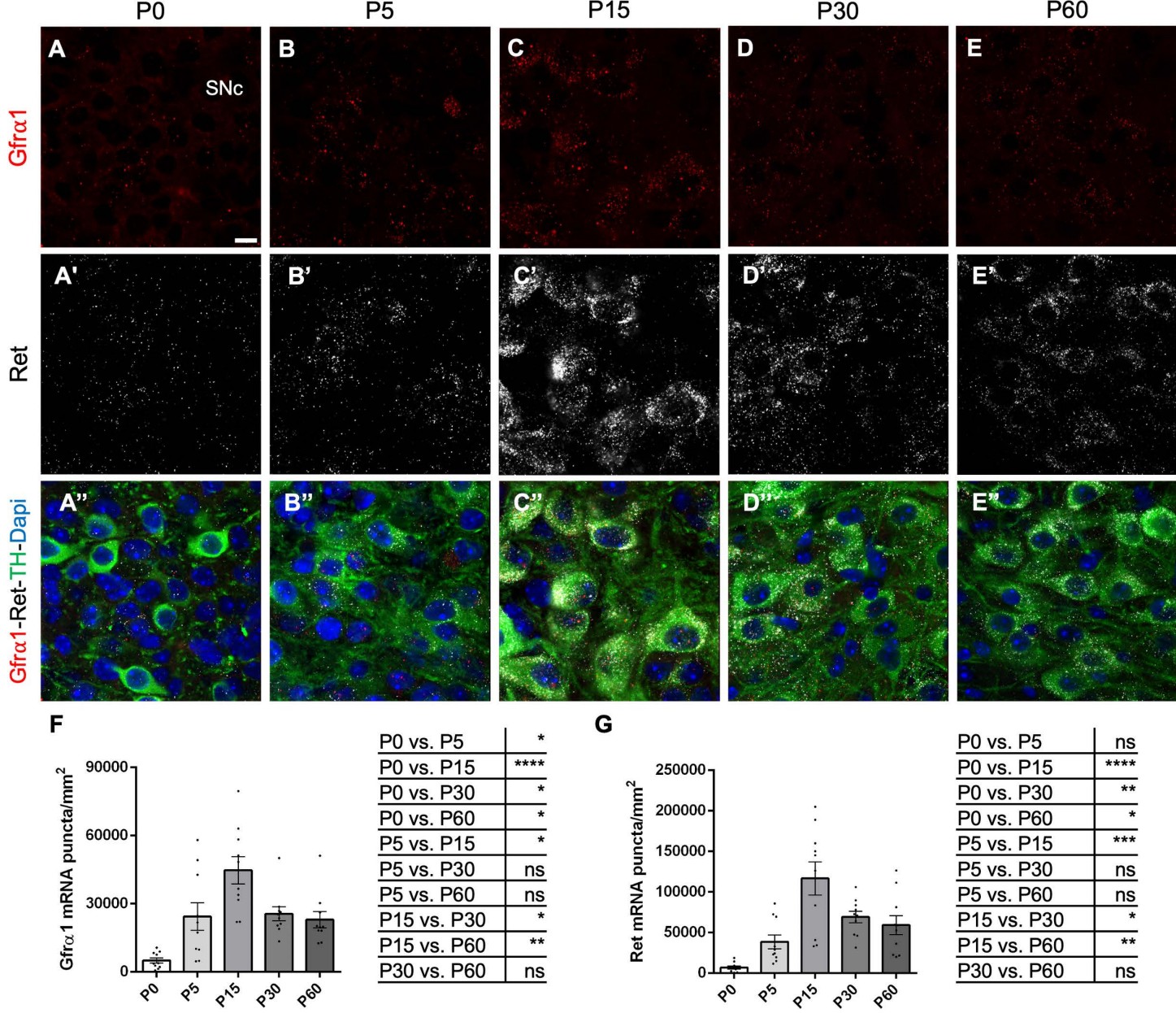

**Fig 1. Developmental expression of *Gfra 1* mRNA in the postnatal SN and dSTR.** (A-E") Representative confocal images of the substantia nigra pars compacta (SNc) showing RNAscope mRNA signal for *Gfra1* (red) and *Ret* (white), immunolabeling for TH (green), and counterstaining with DAPI in coronal brain sections from C57BL6/J mouse at P0 (A), P5 (B), P15(C), P30 (D) or P60 (E). Scale bar = 10 μm. (F-G) Expression levels (± SEM) of *Gfra1* (F) and *Ret* (G) mRNAs quantified by RNAscope puncta analysis in the SNc across postnatal developmental ages. Statistical comparisons among ages are described by the respective table beside. N = 10 SN from 5 mice for all ages. *p < 0,05; **p < 0,01; ***p < 0,001; ****p < 0,0001; One-way Anova.

these cells in the adult SN. The majority of dTOM+ cells in the adult SN of heterozygous mice, approximately 90%, were also TH+ (Fig 3E), indicating that most if not all GFRα1-expressing neurons in this structure are dopaminergic. The proportion of TH+ cells among the remaining dTOM+ population in the null mutants was not changed (Fig 3E). Approximately 60% of TH+ cells in the adult SN also expressed GFRα1 in heterozygous mice, and about a quarter of those cells were lost in the null

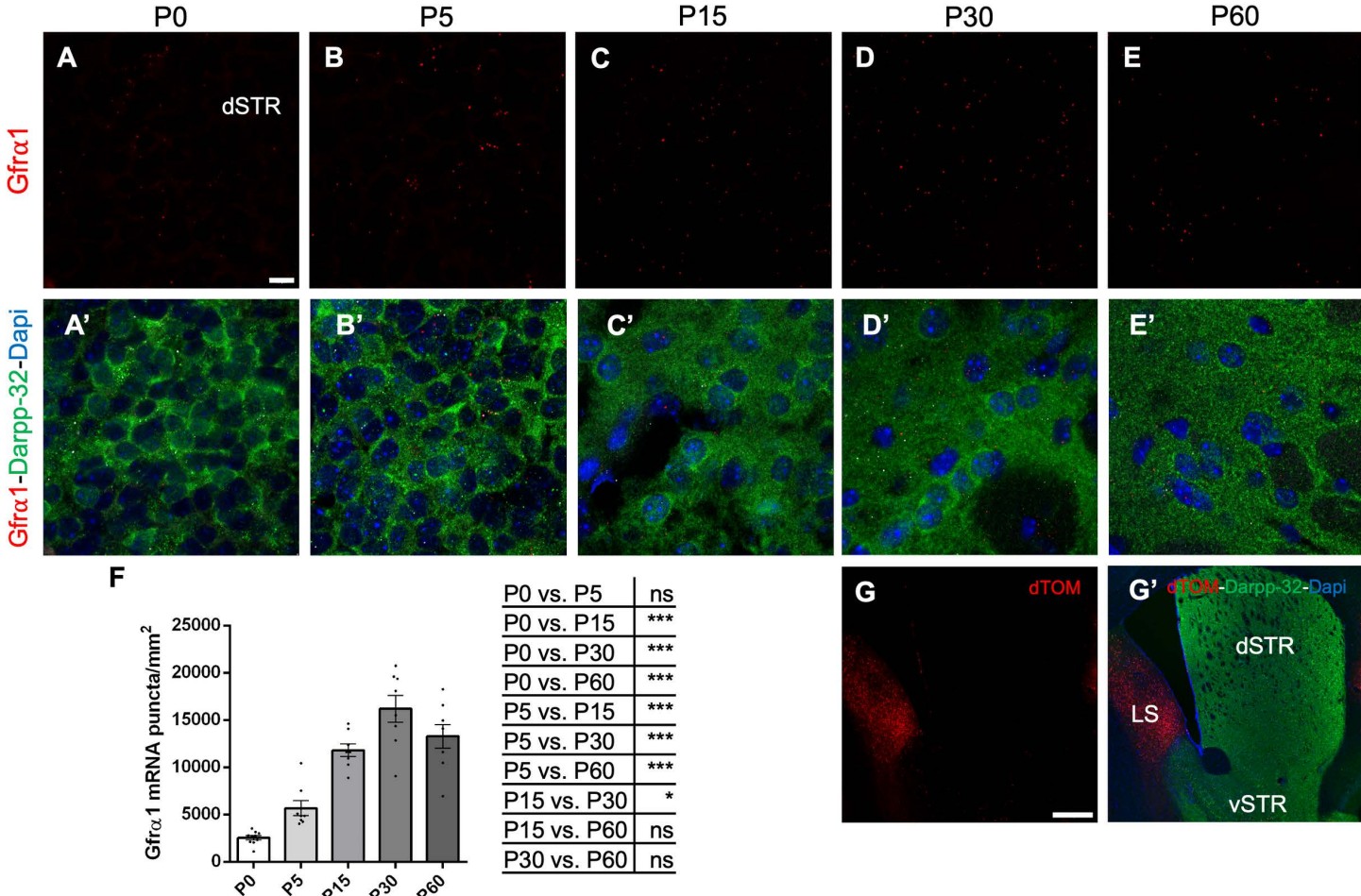

**Fig 2. Sparse *Gfra1* mRNA expression in postnatal dSTR likely arises from SN terminals.** (A-E') Representative confocal images of the dSTR showing RNAscope mRNA signal for *Gfra1* (red), immunolabeling for DARPP-32 (green), and counterstaining with DAPI (blue) in coronal brain sections from C57BL6/J mouse at P0 (A), P5 (B), P15(C), P30 (D) or P60 (E). Scale bar = 10 μm. (F) Expression levels (± SEM) of *Gfra1* (N) mRNA quantified by RNAscope puncta analysis in the dSTR across postnatal developmental ages. Statistical comparisons among ages are described by the table beside. N = 8 dSTR from 4 mice for all ages except P0 (N = 10 dSTR from 5 mice). *p < 0,05; ***p < 0,001; One-way Anova. (G-G') Representative confocal low magnification images of the striatum showing dTomato epifluorescence (red), immunolabeling for DARPP-32 (green) and counterstaining with DAPI (blue) in coronal brain sections of *Gfra1*^CreERT2/+;*Rosa26*^dTOM (GFRα1+/-) mouse injected with tamoxifen at 3 months. GFRα1 dTom+ cells can be observed predominantly in the lateral septum (LS) and sparsely in the ventral striatum (vSTR) but not in the dorsal striatum (dSTR). Scale bar = 400 μm.

mutants (Fig 3F). In order to determine whether this was due to loss of GFRα1 expression or loss of TH+ neurons in the null mutants, we quantified the number of TH+ cells in heterozygous, null, and wild type mice. A reduction of similar magnitude to that observed among dTOM+ cells (approximately 30 cells/mm²) was also found in the TH+ cell population of null mice (Fig 3G). Interestingly, the density of TH+ cells in the adult SN of wild type and *Gfra1* heterozygous mice was indistinguishable (Fig 3G), indicating that the *Gfra1* gene is not haplo-insufficient in dopaminergic neurons, as GFRα1 expression from only one allele was sufficient to support the full complement of TH+ cells in the adult SN. As the loss of TH+ cells was detected only 3 days after deletion of *Gfra1*, we conclude that sustained expression of this receptor is necessary for the maintenance of the full complement of dopaminergic neurons in the adult SN.

Previous studies arrived at contradictory results regarding the requirement of GDNF expression for the maintenance of adult dopaminergic neurons. While Pascual et al. [19] initially reported that GDNF is indispensable for adult

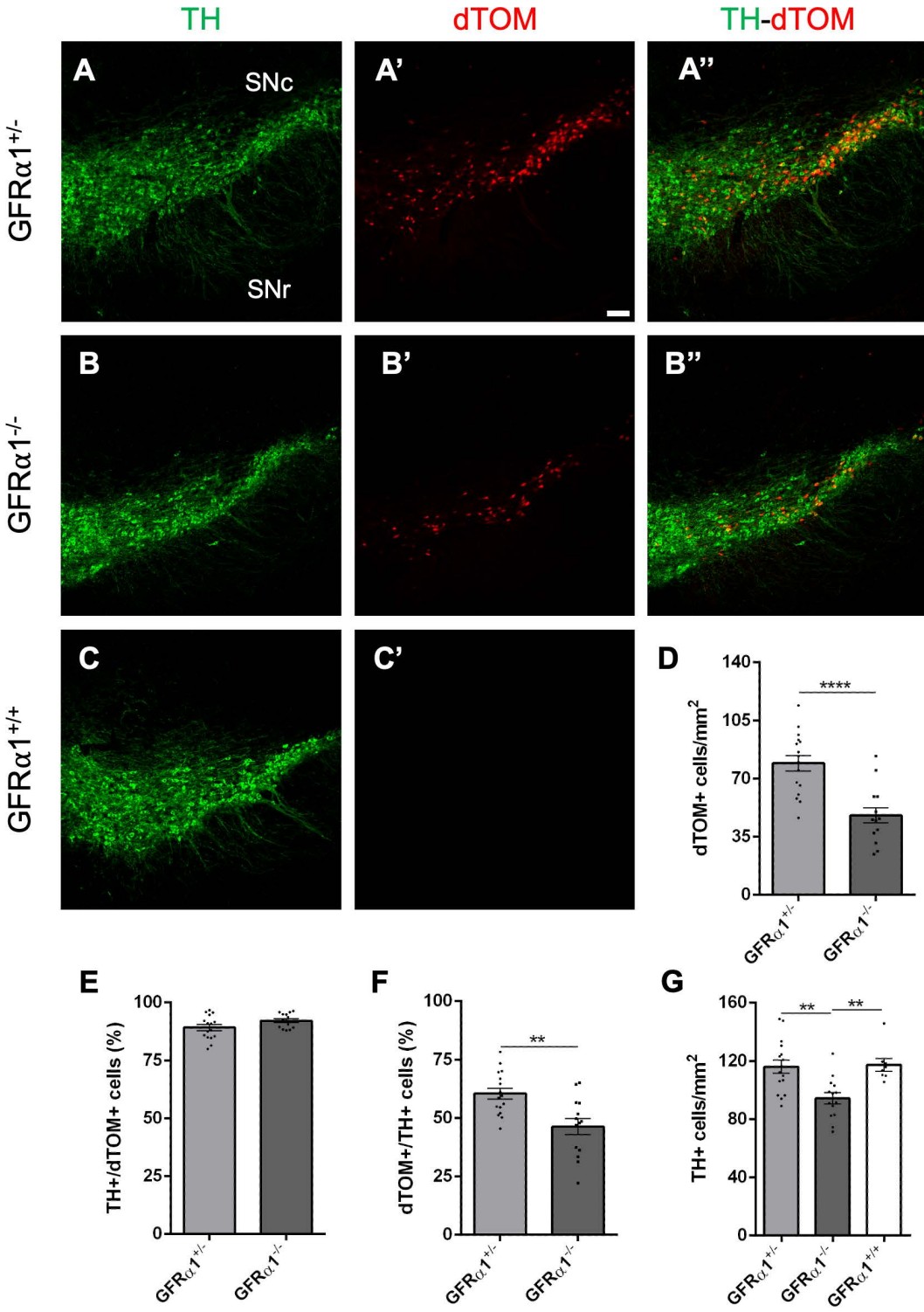

**Fig 3. GFRα1 is necessary for the maintenance of dopaminergic neurons in the adult SN.** (A-C') Representative confocal images of the SN showing TH immunolabeling (green) and dTomato epifluorescence (red) in coronal brain sections of *Gfra1*CreERT2/+;*Rosa26*dTOM (GFRα1+/-, A), *Gfra1*CreERT2/flox;*Rosa26*dTOM (GFRα1-/-, B) or C57BL6/J mouse (GFRα1+/+) injected with tamoxifen at 3 months. Scale bar = 100μm. (D-F) Quantification (± SEM) of GFRα1 dTom+ cells (D), % of GFRα1 dTom+ cells co-labelled with TH (E) and % of TH+ cells co-labelled with dTom (F) in the SN of GFRα1+/- and GFRα1-/- mice. N = 16 and 14 SN from 8 and 7 mice (GFRα1+/- and GFRα1-/-, respectively). **p < 0,01; ****P < 0,0001; T-test. (G) Quantification (± SEM) of TH+ cells in the SN of GFRα1+/-, GFRα1-/- and GFRα1+/+ mice. N = 16, 14 and 8 SN from 8, 7 and 4 mice (GFRα1+/-, GFRα1-/- and GFRα1+/+, respectively). **p < 0,01; One-way Anova.

catecholaminergic neuron survival (see also [29]), Kopra et al. [21] challenged this notion using both Cre-expressing transgenic lines, as in the Pascual et al. study, as well as intrastriatal injection of viruses expressing Cre protein. They reported no loss of SN dopaminergic neurons up to 19 months of age [21]. In an accompanying rebuttal, Pascual et al. pointed to potential problems in the Cre-driven transgenic lines and the relatively inefficient recombination obtained with the viral approach used in the study by Kopra et al. [20]. As mentioned earlier, while conditional ablation of RET did not result in deficits in SN dopaminergic neurons up to 9 months [22], approximately 30% dopaminergic neuron loss was observed in 1 and 2 years old conditional mutant mice [23]. In our present study, we found a 40% loss of dTOM+ neurons in the SN of 3-month-old mice only 3 days after the last tamoxifen injection (7 days after the first). As the majority of dTOM+ cells also expressed TH, but only 60% of TH-expressing neurons were also dTOM+, we believe the loss of SN dopaminergic neurons in these mice to be in the range of 24%, which is in agreement with the TH+ cell counts in this structure across the different genotypes (Fig 3G). Although our results would appear to be more in line with those of Pascual et al., we note that GFRα1 is able to bind other ligands of the GDNF family besides GDNF, notably Neurturin [2,7]. Expression of mRNA encoding Neurturin has been reported in the adult STR at low levels, though similar to *Gdnf* mRNA [30].

## Ablation of GFRα1 preferentially affects SN dopaminergic neurons projecting to the STR

The effect of acute deletion of *Gfra1* on the innervation of the dSTR by adult SN dopaminergic neurons was assessed in heterozygous (GFRα1[+/-]) and null (GFRα1[-/-]) 3-month-old mice by FluoroGold (FLG) tracing followed by tamoxifen injection (Fig 4A). In concordance with our previous results, there was a reduction in FLG+ cells in the SN of null mice 3 days after the last tamoxifen injection (Fig 4B), suggesting decreased innervation of dSTR after acute loss of GFRα1 in SN neurons. FLG tracing was combined with TH immunohistochemistry and dTOM fluorescence (reflecting GFRα1-expressing cells) to assess the impact of *Gfra1* deletion on the ability of cells expressing either or both proteins to innervate the dSTR (Fig 4C–4E'''). In line with our previous results, the proportion of FLG+ cells was reduced among the dTOM+ subpopulation, both TH+ (the majority) and TH-, in the SN of *Gfra1* null mice compared to GFRα1[+/-] heterozygous (Fig 4F). Approximately 50% of SN dTOM+ cells could be labeled by FLG tracing from the dSTR (Fig 4G), and dTOM+ cells were most predominantly reduced among the FLG+ subpopulation, both TH+ and TH- (Fig 4G). In line with this, a significant decrease was observed in the proportion of TH+/dTOM+ cells that innervated the dSTR (FLG+) but not among those that did not (FLG-) in the null mutants compared to GFRα1[+/-] heterozygous (Fig 4H). These results suggest that acute disruption of GFRα1 expression preferentially affects SN dopaminergic neurons projecting to the STR. Low but measurable levels of both *Gdnf* and *Neurturin* mRNAs, encoding the two main GFRα1 ligands, have been found in the striatum [30], suggesting that the subpopulation of GFRα1-expressing neurons in the SN most critically dependent on continued expression of this receptor are those reaching potential sources of GFRα1 ligands. The resilience of GFRα1-expressing SN neurons that do not project to the STR may be due to alternative sources of trophic support, such as that provided by BDNF/TrkB signaling.

In summary, this study shows that an important fraction of GFRα1-expressing SN neurons projecting to the dSTR are critically dependent on the continued expression of this receptor. The possible roles of other components of this neurotrophic system in such dependency, including GDNF, Neurturin, RET and NCAM, remain to be thoroughly assessed. Interestingly, this latter molecule has been implicated in dopaminergic neuron survival [31,32], but whether such activity involves GFRa1 has not yet been determined. Moreover, although here we favor a cell-autonomous requirement of GFRα1 in SN neurons, we note that tamoxifen injection in *Gfra1*[CreERT2/flox];*Rosa26*[dTOM] mice results in the loss of GFRα1 in all brain cells that normally express the receptor. Thus, we cannot at present exclude the possibility that deletion of GFRα1 in brain areas other than the SN may have affected the survival of SN dopaminergic cells in a non-cell-autonomous fashion, such as the cholinergic nicotinic neurotransmission from the pedunculopontine nucleus [33].

Finally, these results highlight once again the importance of investigating the actions of GFRα1 and its ligands in the pre-clinical context of PD pathology. PD becomes symptomatic only after more than 50% of striatal dopamine levels and

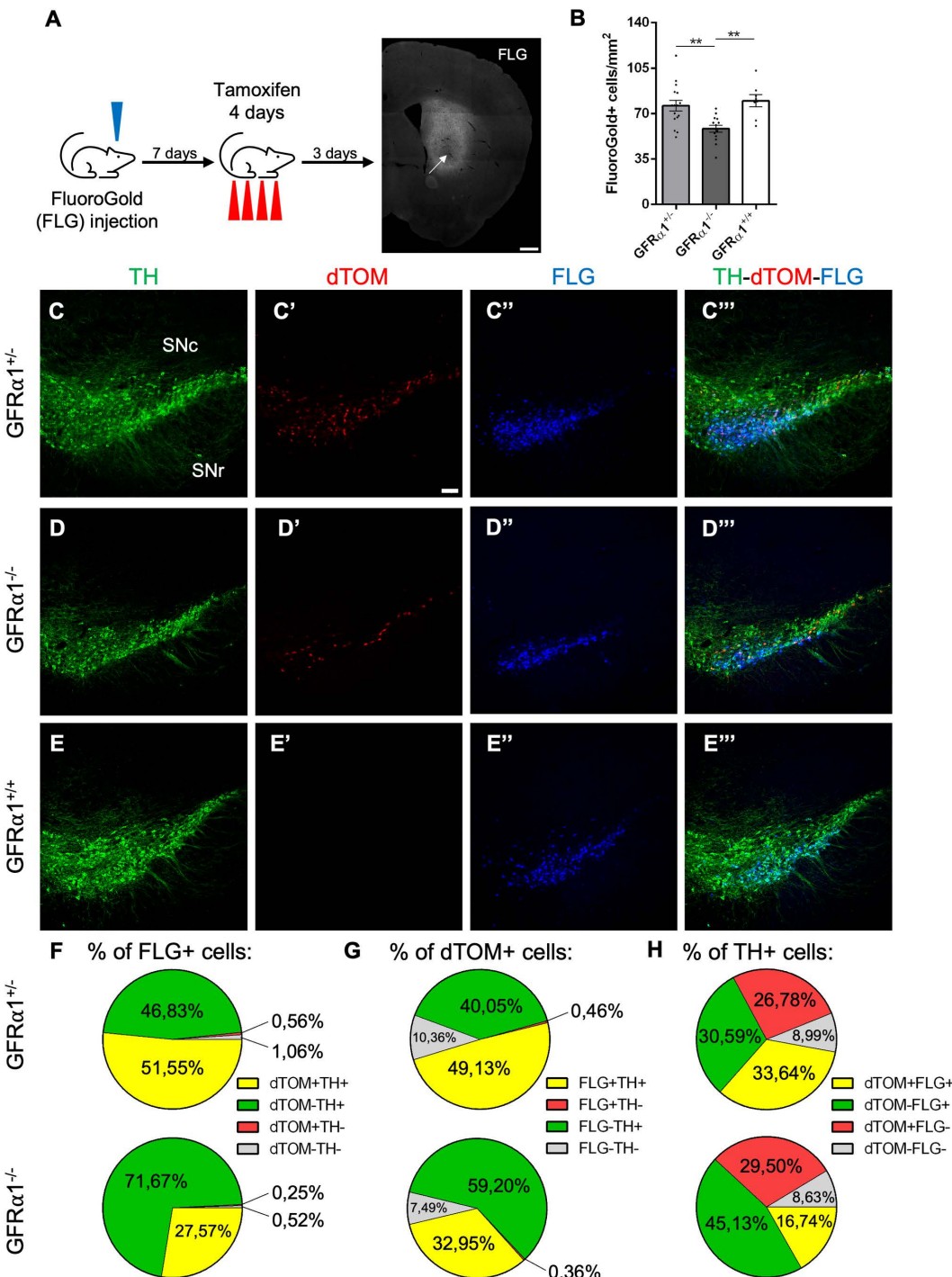

**Fig 4. Ablation of GFRα 1 preferentially affects SN dopaminergic neurons projecting to the STR.** (A) Schematic of the protocol used for tracing analysis of GFRα1 neurons in the SN. GFRα1+/-, GFRα1-/- and GFRα1+/+ mice were stereotaxically injected with FLG retrograde tracer in the dSTR. One week later, mice were placed on a daily tamoxifen administration regime for four days. Three days after the last administration, mice were histologically analyzed. FLG injection site (arrow) can be observed by its blue epifluorescence in a confocal coronal reconstruction of the dSTR. Scale bar = 500 μm. (B) Quantification (± SEM) of FLG+ cells in the SN of GFRα1+/-, GFRα1-/- and GFRα1+/+ mice. N = 16, 14 and 8 SN from 8, 7 and 4 mice (GFRα1+/-, GFRα1-/- and GFRα1+/+, respectively). **p < 0,01; One-way Anova. (C-E''') Representative confocal images of the SN showing TH immunolabeling (green), dTomato (red) and FLG (blue) epifluorescence in coronal brain sections of GFRα1+/- (C), GFRα1-/- (D) or GFRα1+/+ (E) mice injected with tamoxifen at 3 months. Scale bar = 100μm.

(F-H) Characterization profile of FLG+ cells (F), GFRα1 dTom+ cells (G) and TH+ cells (H) in the SN of injected GFRα1+/- (upper panels) and GFRα1-/- (lower panels) mice. Numbers show the % of cells for each combination of markers (FLG, GFRα1 and TH).

30–40% of SNpc neurons are lost (reviewed in [18]). In the present study, we have observed a reduction of 24% in the complement of SN dopaminergic neuron that projects to the STR following GFRα1 deletion. Therefore, these mice may be just below the threshold for manifestation of classical PD behavioral abnormalities, a possibility that will be the subject of future studies. Futhermore, the inconclusiveness of previous clinical trials has raised concerns about aberrant sprouting at the site of GDNF administration and GDNF dosing, which usually exceeds endogenous GDNF levels (reviewed in [18]).

One aspect that was not investigated in the previous clinical trials is whether these patients still present available receptors for GDNF to exert its trophic effects. In a rat model of PD, it was found that GFRα1 levels decrease significatnly in the STR and SN after PD induction by injection of 6-hydroxydopamine and correlate with nigrostriatal neuronal loss [34]. The same report also observed an increase of GFRα1 coming from GFAP+ astrocytes, possibly as a mechanism to counteract the loss of GFRα1+SN neurons in the PD model. Since GFRα1 in its soluble form can be expressed and secreted by glial cells [14], astrocytic GFRα1 *trans*-signaling might prove to be a novel therapeutic target to provide trophic support on adjacent remaining SN neurons in PD. Further investigation into the expression and activation of GFRα1 in the nigro-striatal pathway is necessary to provide solutions to the issues observed in the clinical trials.

## Supporting information

**S1 File. Manuscript raw data.** The raw data used to generate all bar graphs in this study are found in this file. (XLSX)

## Acknowledgments

The authors would like to thank Mart Saarma and Jaan-Olle Andressoo (University of Helsinki, Finland) for providing *Gfra-1*flox mice.

## Author contributions

**Conceptualization:** Alvaro Carrier-Ruiz, Annika Andersson, Diana Fernández-Suárez.

**Formal analysis:** Alvaro Carrier-Ruiz.

**Funding acquisition:** Carlos F. Ibáñez.

**Investigation:** Alvaro Carrier-Ruiz, Annika Andersson, Diana Fernández-Suárez.

**Methodology:** Alvaro Carrier-Ruiz, Annika Andersson, Diana Fernández-Suárez.

**Project administration:** Carlos F. Ibáñez.

**Supervision:** Carlos F. Ibáñez.

**Visualization:** Alvaro Carrier-Ruiz, Annika Andersson, Diana Fernández-Suárez.

**Writing – original draft:** Alvaro Carrier-Ruiz, Carlos F. Ibáñez.

**Writing – review & editing:** Alvaro Carrier-Ruiz, Annika Andersson, Diana Fernández-Suárez, Carlos F. Ibáñez.

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
