## [Decision Letter · Decision Letter 0]

4 Jul 2025

PONE-D-25-26528GDNF receptor GFRα1 is necessary for the maintenance of dopaminergic neurons in the adult substantia nigraPLOS ONE

Dear Dr. Carrier Ruiz,

Thank you for submitting your manuscript to PLOS ONE. After careful consideration, we feel that it has merit but does not fully meet PLOS ONE’s publication criteria as it currently stands. Therefore, we invite you to submit a revised version of the manuscript that addresses the points raised during the review process.

This is an important study to advance our understanding of GDNF signaling and the critical importance of GDNF receptors; a much-overlooked component.  The authors did a good job interrogating the underappreciated role of GFR-a1 in regulating nigrostriatal neuron viability.  The study shows the critical importance of GFR-a1 in nigrostriatal dopamine neuron viability.   It is this editor's opinion that the impact of this important study can increase by application of the findings in context of Parkinson's disease.  Whereas much attention has focused upon GDNF itself (the ligand) and its impact on nigrostriatal function, far less attention has been placed upon GFR-a1 and RET and the role these receptors play in mediating GDNF signaling.  That said there are at least a few studies that show that the authors main findings are applicable in disease states (Parkinson's for example).  Kasanga and colleagues (PMID 37178997) showed that GFR-a1 expression was highly correlative to TH neuron expression during toxin-driven loss of nigrostriatal neurons. Pruett and Salvatore showed that the soluble form of GFR (PMID 11182089) could increase TH and DA levels in the SN (PMID 23321789).  Thus, with the authors showing the critical dependence of GFR-a1 on nigrostriatal neuron viability with their intricate and well-controlled study, their result has direct bearing upon Parkinson's disease and that expression of this receptor may be a critical limiting factor in the success of any GDNF therapy.

We look forward to receiving your revised manuscript.

Kind regards,

Michael Francis Salvatore

Academic Editor

PLOS ONE

2. To comply with PLOS ONE submissions requirements, in your Methods section, please provide additional information regarding the experiments involving animals and ensure you have included details on (1) methods of sacrifice, and (2) efforts to alleviate suffering.

[The authors would like to thank Mart Saarma and Jaan-Olle Andressoo (University of Helsinki, Finland) for providing  Gfra1flox mice. This research was funded by grants to C. F. I. from the Swedish Research Council (2020-01923).]

 [C. F. I.

Grant number: 2020-01923

Swedish Research Council https://www.vr.se/english.html.

The funders did not play any role in the study design, data collection and analysis, decision to publish, or preparation of the manuscript.]

4. In the online submission form, you indicated that [All relevant data are within the manuscript, otherwise are available on request from the author.].

Additional Editor Comments:

This is an important study to advance our understanding of GDNF signaling and the critical importance of GDNF receptors; a much-overlooked component. The authors did a good job interrogating the underappreciated role of GFR-a1 in regulating nigrostriatal neuron viability. The study shows the critical importance of GFR-a1 in nigrostriatal dopamine neuron viability. It is this editor's opinion that the impact of this important study can increase by application of the findings in context of Parkinson's disease. Whereas much attention has focused upon GDNF itself (the ligand) and its impact on nigrostriatal function, far less attention has been placed upon GFR-a1 and RET and the role these receptors play in mediating GDNF signaling. That said there are at least a few studies that show that the authors main findings are applicable in disease states (Parkinson's for example). Kasanga and colleagues (PMID 37178997) showed that GFR-a1 expression was highly correlative to TH neuron expression during toxin-driven loss of nigrostriatal neurons. Pruett and Salvatore showed that the soluble form of GFR (PMID 11182089) could increase TH and DA levels in the SN (PMID 23321789). Thus, with the authors showing the critical dependence of GFR-a1 on nigrostriatal neuron viability with their intricate and well-controlled study, their result has direct bearing upon Parkinson's disease and that expression of this receptor may be a critical limiting factor in the success of any GDNF therapy.

Reviewers' comments:

Reviewer's Responses to Questions

**Comments to the Author**

1. Is the manuscript technically sound, and do the data support the conclusions?

Reviewer #1: Yes

Reviewer #2: Yes

Reviewer #3: Yes

2. Has the statistical analysis been performed appropriately and rigorously? 

Reviewer #1: Yes

Reviewer #2: Yes

Reviewer #3: Yes

3. Have the authors made all data underlying the findings in their manuscript fully available?

Reviewer #1: Yes

Reviewer #2: No

Reviewer #3: Yes

4. Is the manuscript presented in an intelligible fashion and written in standard English?

Reviewer #1: Yes

Reviewer #2: Yes

Reviewer #3: Yes

5. Review Comments to the Author

Reviewer #1: This is an interesting study that sought to establish if there is a physiological requirement of the GFRα1 component of the neurotrophic system for the continuous maintenance of SN dopaminergic neurons in the adult brain.

Please note a few comments below:

1. In the methods section, please include a description for the mice which were used for the postnatal study (figure 1). The only mention of these mice seems to be in the tissue processing section (page 18; lines 297-299)

2. It is noted that both males and females were used in this study. Were there any sex differences seen in the postnatal mRNA expression levels of Gfra1 and Ret in the mouse SN, postnatal Gfrα1 mRNA expression in the dorsal striatum, or the acute deletion of Gfra1 on adult SN dopaminergic neurons?

3. The authors noted that tamoxifen injection in Gfrα1CreERT2/flox;Rosa26dTOM mice results in the loss of GFRα1 in all brain cells that normally express the receptor. Could the authors cite data highlighting loss of GFRa1 in key brain areas which may affect the survival of SN dopaminergic cells in a non-cell-autonomous fashion from their group or other groups which would help the reader to better put their data in context?

4. The authors suggest that acute disruption of GFRα1 expression preferentially affects SN dopaminergic neurons projecting to the striatum. Based on work done by either this group or others, would (could) this loss result in behavioral effects (signs) mimicking Parkinson’s disease? Could the authors briefly speak to this theme as well?

Minor comments

Please note the following suggested revisions:

1. Correct the typo on Page 10 (Introduction [lines 50-51]): “The glycoprotein Syndecan-3 has also been proposed as a receptor fror GDNF family ligands (13),”

2. Ensure the consistent use of abbreviations throughout the manuscript. For eg, dorsal striatum was defined as “dSTR” on page 11 (line 86). However, the term was repeated multiple times throughout the manuscript with and without the abbreviation.

Reviewer #2: In their manuscript, Ruiz and colleagues utilize tamoxifen-induced tissue specific genetic recombination mouse model to study the effect of Gfra1 ablation on the maintenance of adult nigral dopaminergic neurons. Gfra1 is a co-receptor for GDNF, a neurotrophic factor tested in multiple clinical trials for its disease-modifying effect in Parkinson’s disease patients, with controversial outcomes. There is a long-standing debate in the literature regarding the role of GDNF in the maintenance of adult dopaminergic neurons. This issue has been previously addressed by genetic deletion of GDNF and its receptor Ret in the adult mice. The present study extends this knowledge by carefully evaluating the role of Gfra1 ablation in nigral dopaminergic neurons in the adult mouse. Utilizing a combination of genetic tools to allow fluorescent labeling of Gfra1 knockout neurons and fluorogold tracing, the authors convincingly demonstrate that that Gfra1 is required for the maintenance of a subset of adult nigral dopaminergic neurons, and that the ablation of Gfra1 preferentially affects the subset of neurons that project to the striatum. I have several comments regarding the study, which are listed below.

The authors should carefully describe and discuss limitations of their study. First, their results are obtained in the mouse model and its translation to human dopaminergic neurons is not clear. Indeed, there are multiple studies showing excellent disease modifying effects, including the reduction of aSyn aggregation, of GDNF in the rodent PD models, and yet the results of clinical trials are inconclusive. Second, the authors should highlight potential technical issues. For example, they could not exclude the possibility of incomplete recombination: some dTOM positive cells may still express Gfra1, and some Gfra1-null cells may not be dTOM positive. The authors followed the fate of dopaminergic neurons by TH immunostaining only at a very short timepoint (3 days) after tamoxifen administration. The Gfra1 ablation may thus be just causing a (potentially only transient) down-regulation of TH expression. Fluorogold tracing results suggest the degeneration of neurons, but statistical analysis of these data is not shown, and it is hard to imagine that neurons would degenerate in such a short time frame. Also, potentially more neurons would degenerate at longer timepoints. The study would be more convincing if the authors had examined Gfra1 positive nigral dopaminergic neurons and striatal dopaminergic innervation at multiple timepoints after Gfra1 ablation.

The authors should demonstrate or, if it was shown before, provide a clear comment and reference about potential “leakiness” of CreERT2, by including a mouse group injected with oil+10% ethanol vehicle control.

Bar graphs should show individual datapoints (see https://doi.org/10.1038/s41551-017-0079).

The quality of immunostaining and RNAscope images (at least in the reviewers’ PDF) is very poor, especially red channels on Figs. 1 and 2.

Line 95: “substantial” -> substantia

Line 140: “septus” -> septum

Reviewer #3: In this manuscript, Ruiz et al. seek to answer key questions regarding GDNF signaling and its importance to maintenance of dopamine neurons in the substantia nigra. More specifically, through a series of intricate experiments, the authors attempt to elucidate the role and importance of GFRα1 in mediating GDNF signaling. Their key findings are that GFRα1 and RET mRNA expression peak around postnatal day 15 and stabilize at about half-peak levels from P30 onward. They also find that while there is sparse expression of GFRα1 in the dorsal striatum, there is no expression of dTomato in transgenic mice with a dTomato reporter under the control of GFRα1CreERT2. The authors interpret this as meaning that GFRα1 expression in the dorsal striatum must come from SN dopaminergic axons. Next, the authors assess what the impact of conditional deletion of GFRα1 expression has on SNc neurons finding that it causes an acute reduction in dTOM+ cells, the % of TH+ cells that also express dTOM as well as a total reduction in the number of TH+ cells relative to both a heterozygous deletion and WT mice. Finally, the authors repeat this experiment in mice for whom they have labeled SNc neurons projecting to the striatum using fluorogold tracing and find that GFRα1 ablation decreases FLG staining in the SN especially among dTOM+ cells. Furthermore, in TH+ cells, they find a reduction in dTOM+ cells projecting to the striatum (FLG+) but no change in dTOM+ cells not projecting to the striatum (FLG-). They interpret this to mean that the cells most dependent upon GDNF signaling via GFRα1 are the SN dopamine neurons projecting to the dorsal striatum. On the whole, the experiments presented are well done, the interpretations are reasonable, and this manuscript by Ruiz et al. offers a crucial contribution to the field. Still, I think there are some things that could be addressed to improve the manuscript.

1. It is mentioned that “insertion of CreERT2 in the Gfra1 locus inactivates the targeted allele,” and while there are several references cited, there wasn’t any data presented demonstrating that this does in fact reduce expression of Gfra1. Furthermore, it would be worthwhile to demonstrate in the Gfrα1CreERT2/flox that Gfrα1 expression is truly ablated.

2. One question I had after reading the manuscript is whether ablation of Gfrα1 expression actually leads to loss of neurons or just alters their programing such that they no longer express dopaminergic markers. I think it would be worth using a nondopaminergic marker or simply using DAPI staining to label the nuclei so you can determine if Gfrα1 ablation actually leads to loss neurons or not.

6. PLOS authors have the option to publish the peer review history of their article (what does this mean?). If published, this will include your full peer review and any attached files.

Reviewer #1: No

Reviewer #2: **Yes: **Andrii Domanskyi

Reviewer #3: **Yes: **Brandon Scott Pruett

---

## [Author Response · Author response to Decision Letter 1]

16 Jul 2025

We thank the editor and the reviewers for their valuable comments. We have used your input to make the requested changes to the manuscript. Please find in the submitted "Response to Reviewers_Carrier-Ruiz" file the answers to each point raised by the editor and reviewers. We believe that the changes made have improved the manuscript which we now hope will be acceptable for publication in PLOS ONE.

Sincerely,

Alvaro Carrier Ruiz, PhD

---

## [Decision Letter · Decision Letter 1]

6 Aug 2025

PONE-D-25-26528R1GDNF receptor GFRα1 is necessary for the maintenance of dopaminergic neurons in the adult substantia nigraPLOS ONE

Dear Dr. Carrier Ruiz,

Thank you for submitting your manuscript to PLOS ONE. After careful consideration, we feel that it has merit but does not fully meet PLOS ONE’s publication criteria as it currently stands. Therefore, we invite you to submit a revised version of the manuscript that addresses the points raised during the review process.

We look forward to receiving your revised manuscript.

Kind regards,

Michael Francis Salvatore

Academic Editor

PLOS ONE

Journal Requirements:

Additional Editor Comments:

The authors rightly connect their work to Parkinson’s disease, given what has been revealed with preclinical and clinical GDNF studies. GFR-α1 function in dopamine neurons has been far less studied, but arguably just as important as its cognate ligand. Again, I refer the authors to consider comments from reviewer 1 in points 3 and 4 from the first submission regarding GFR-α1 expression in the nigrostriatal pathway and potential application to Parkinson’s. Kasanga et al (Exp Neurol, 2023) reported progressive GFRα1 loss that was in association with loss of nigrostriatal neurons. This loss was specific to nigrostriatal neurons. Moreover, GFRα1 was upregulated in astrocytes as the lesion progression continued. This finding seems to be very applicable to this study and may be useful in the discussion.

Reviewer's Responses to Questions

**Comments to the Author**

1. If the authors have adequately addressed your comments raised in a previous round of review and you feel that this manuscript is now acceptable for publication, you may indicate that here to bypass the “Comments to the Author” section, enter your conflict of interest statement in the “Confidential to Editor” section, and submit your "Accept" recommendation.

Reviewer #1: All comments have been addressed

Reviewer #2: All comments have been addressed

Reviewer #3: All comments have been addressed

2. Is the manuscript technically sound, and do the data support the conclusions?

Reviewer #1: Yes

Reviewer #2: Yes

Reviewer #3: Yes

3. Has the statistical analysis been performed appropriately and rigorously? 

Reviewer #1: Yes

Reviewer #2: Yes

Reviewer #3: Yes

4. Have the authors made all data underlying the findings in their manuscript fully available?

Reviewer #1: Yes

Reviewer #2: Yes

Reviewer #3: Yes

5. Is the manuscript presented in an intelligible fashion and written in standard English?

Reviewer #1: Yes

Reviewer #2: Yes

Reviewer #3: Yes

6. Review Comments to the Author

Reviewer #1: (No Response)

Reviewer #2: (No Response)

Reviewer #3: For the most part, the authors have thoughtfully addressed the concerns raised by the reviewers, and the result is an improved manuscript.

7. PLOS authors have the option to publish the peer review history of their article (what does this mean?). If published, this will include your full peer review and any attached files.

Reviewer #1: No

Reviewer #2: **Yes: **Andrii Domanskyi

Reviewer #3: **Yes: **Brandon Scott Pruett, MD, PhD

---

## [Author Response · Author response to Decision Letter 2]

13 Aug 2025

Dear Dr. Michael Francis Salvatore,

We thank you and the reviewers for their valuable comments. We are pleased to have satisfactorily addressed all the reviewers’ comments, resulting in an improved manuscript. We have used your new input to make the requested changes to the manuscript. Please find in the submitted file the response to your raised point. We believe that the changes made have refined the manuscript, which we now hope will be acceptable for publication in PLOS ONE.

Sincerely,

Alvaro Carrier Ruiz, PhD

---

## [Editor Report · Decision Letter 2]

15 Aug 2025

GDNF receptor GFRα1 is necessary for the maintenance of dopaminergic neurons in the adult substantia nigra

PONE-D-25-26528R2

Dear Dr. Carrier Ruiz,

We’re pleased to inform you that your manuscript has been judged scientifically suitable for publication and will be formally accepted for publication once it meets all outstanding technical requirements.

Kind regards,

Michael Francis Salvatore

Academic Editor

PLOS ONE

Additional Editor Comments (optional):

This study help will provide a necessary and critical piece of the puzzle if we are to effectively treat Parkinson's disease with GDNF. Outstanding study and manuscript.
---

## [Editor Report · Acceptance letter]

PONE-D-25-26528R2

PLOS ONE

Dear Dr. Carrier Ruiz,

I'm pleased to inform you that your manuscript has been deemed suitable for publication in PLOS ONE. Congratulations! Your manuscript is now being handed over to our production team.

Kind regards,

on behalf of

Dr. Michael Francis Salvatore

Academic Editor

PLOS ONE